# Waste Electrical and Electronic Equipment Reutilization in China

**Rong Wang [1], Yi Deng [2], Shuyuan Li [2], Keli Yu [3], Yi Liu [4], Min Shang [4], Jiqin Wang [1], Jiancheng Shu [1], Zhi Sun [5], Mengjun Chen [1,*] and Qian Liang [1]**

1   Key Laboratory of Solid Waste Treatment and Resource Recycle, Ministry of Education, Southwest University of Science and Technology, Mianyang 621010, China; wr276432218@163.com (R.W.); wjq_swust@163.com (J.W.); sjcees@126.com (J.S.); lq_swust@163.com (Q.L.)
2   Solid Waste and Chemical Management Technology Center of the Ministry of Ecological Environment, Beijing 100037, China; dengyi@meescc.cn (Y.D.); lishuyuan@meescc.cn (S.L.)
3   China National Resources Recycling Association, Beijing 100037, China; ykl@crra.com.cn
4   Sichuan Solid Waste and Chemicals Management Center, Chengdu 610000, China; Y.liu@163.com (Y.L.); shangmin@ruc.edu.cn (M.S.)
5   Institute of Process Engineering, Chinese Academy of Sciences, Beijing 100190, China; sunzhi@ipe.ac.cn
*   Correspondence: kyling@swust.edu.cn

**Abstract:** Waste electrical and electronic equipment (WEEE), also called electronic waste or e-waste, the core of "urban mining", is attracting more and more attention to its pollution control and circular recycling. Hence, we defined WEEE, preliminarily discussed its history in China and pointed out that China has made great achievements in WEEE circular reutilization and pollution control. Meanwhile, we analyzed the four levels of circular WEEE recycling: repair, reuse and remanufacture, waste-to-materials, waste-to-products and waste-to-energy, and also put forward questions during this process. Moving forward, WEEE management will turn to intelligent management targeted on hazardous waste and other pollution, not merely the guidelines. Meanwhile, WEEE technology will transfer to value-added and automated reutilization, not just simple dismantling.

**Keywords:** WEEE; fund; circular reutilization; pollution management; intelligent management; value-added and automated reutilization

## 1. Introduction

E-waste, or electronic waste, with a formal name of waste electric and electronic equipment (WEEE), is attracting wide attention in both China and around the world. As the fastest growing waste stream in the world, it is the main source of secondary raw materials of resources, such as Cu and precious metals like Pd, Pt, Au and Ag [1]. From an economic point of view, WEEE has a higher economic value than the original ores [2]. However, it also features potential hazards to the environment and human health due to the high concentrations of heavy metals and organic pollutants [3]. The diversity of the constituent materials, high-value but hazardous, make the recycling of WEEE a complicated and demanding process. Since "the Guiyu issue" of 2002, a series of successes have been achieved because both the industry and the management of circular e-waste recycling have undergone huge development. However, it still necessary to emphasize the concept of the ordinary people who dismantle and use WEEE in individual workshops to improve the management system and, most importantly, break and improve the simple and original e-waste recycling scene and further innovate the e-waste recycling technologies with the aim of supporting the e-waste recycling industry and management. Therefore, to improve the understanding of e-waste by ordinary people, we analyzed the development of e-waste in China, pointed out the achievements in circular e-waste recycling and pollution control from the concept of e-waste. Then, we discussed the problems of the e-waste recycling industry and management in detail on the basis of the four levels of circular solid

waste reutilization: repair/reuse/remanufacture, waste-to-materials, waste-to-products and waste-to-energy. Finally, we pointed out the direction of the e-waste recycling industry based on the industry ecology or the integrated solid waste management strategy.

## 2. Definition

The definition of WEEE, also known as e-waste, electronic waste, is slightly different across countries and associations, and its official name in China has also changed several times. The official name of WEEE in China changed few times though its definition slightly varied. In 2003, it was called "electronic waste" in the "Announcement of the State Environmental Protection Administration on Strengthening the Environmental Management of Electronic Waste" [4]. Its definition was specified in 2007 by the "Administrative Measures for the Prevention and Control of Environmental Pollution by Electronic Waste": the discarded electronic and electrical products or electronic and electrical equipment, the discarded parts and components thereof, as well as the articles and substances that are subject to the management of electronic waste as prescribed by the State Environmental Protection Administration together with other relevant departments, including the obsolete products or equipment generated in the industrial production, the obsolete semi-finished products and residues, the obsolete products generated in the repair, renovation and reproduction of products or equipment, the products or equipment discarded in the daily life or in the activities of providing services for daily life, as well as the products or equipment prohibited to be produced or imported by any law or regulation [5]. Then, in 2009, after the promulgation of the "Regulation on the Administration of the Recovery and Disposal of Waste Electrical and Electronic Products" (revised in 2019), its official name was harmonized with the international one [6]. In 2010, its definition was further clarified as follows: electrical and electronic equipment that the owner of the product no longer uses and has discarded or abandoned (including all parts/components, elements/devices and materials), as well as the substandard products, scrap products and expired products generated during the production, transportation and sales processes ("Technical Specifications for Pollution Control in the Treatment of Waste Electrical and Electronic Products" (HJ527-2010)) [7].

## 3. Chinese WEEE Management Policy

On 25 February 2002, the Basel Action Network (BAN) and the Silicon Valley Toxics Coalition (SVTC) issued a long report "Exporting Harm: The High-Tech Trashing of Asia" and pointed out that 80% of the world's e-waste is transferred to Asia, 90% of which goes to China [8]. Guiyu, China, has suffered an extremely serious ecological and environmental crisis due to the recycling of non-ferrous metals and precious metals from WEEE using primitive disposal methods, known as the Chinese "first village of e-waste dismantling". In Guiyu, primitive and crude techniques were used, including (1) dismantling electronic equipment; (2) heating and manually removing components from printed circuit boards; (3) opening burnt cables and wires to recover metals; (4) chipping and melting plastics; (5) toner sweeping; (6) open acid leaching of e-waste to recover precious metals. These techniques released a large quantity of toxic heavy metals and organic pollutants into the workplace and the surrounding environment [9]. In view of this, on 3 July 2002, the Ministry of Foreign Trade and Economic Cooperation, the General Administration of Customs and the State Environmental Protection Administration of the People's Republic of China jointly announced the "Catalog of Prohibited Imports" (fifth batch, Announcement No. 25, 2002) [10] and prohibited the import of 21 kinds of WEEE (including its parts, disassembled parts, broken parts and smashed parts unless otherwise specified by the state). On 26 August 2003, the State Administration of Environmental Protection issued the "Announcement of the State Environmental Protection Administration on Strengthening the Environmental Management of Electronic Waste". This is the first enacted regulation in China about WEEE, the first landmark of the Chinese electronic waste industry, declaring that WEEE management in China has entered a stage of rapid development from the initial stage.

Since then, China has promulgated the "Technical Policy for Pollution Prevention and Control of Waste Household Appliances and Electronic Products" (2006) [11], "Measures for the Control of Pollution from Electronic Information Products" (2007) [12], "Administrative Measures for the Recovery of Renewable Resources" (issued in 2007 and revised in 2019) [13], "Administrative Measures for the Prevention and Control of Environmental Pollution by Electronic Waste" (2008) [5], "Technical Specifications for Pollution Control in the Treatment of Waste Electrical and Electronic Products" (HJ527-2010) and other relevant laws and regulations one after another to guide the development of the Chinese WEEE industry. However, the most important and well-respected "Regulation on the Administration of the Recovery and Disposal of Waste Electrical and Electronic Equipment" was officially announced on 25 February 2009 and formally implemented on 1 January 2011 (revised in 2019) [14], nearly ten years from incubation to publication and then to implementation. It established a multichannel recycling and centralized processing system, a qualification licensing system, and required the establishment of a WEEE processing fund, known as the Chinese version of the WEEE Directive. The formal implementation of this regulation is another landmark event in the Chinese e-waste industry, announcing that the Chinese WEEE management has entered a stage of gradual improvement from rapid development.

In order to coordinate and improve the implementation of the "Regulation on the Administration of the Recovery and Disposal of Waste Electrical and Electronic Equipment", many laws or regulations, such as "Measures for the Administration of Recycling of Waste Electrical and Electronic Products (Draft for Solicitation of Comments)" [15], "Measures for the Collection, Use and Administration of Funds for the Disposal of Waste Electrical and Electronic Equipment" [16], "Notice on Further Clarifying the Scope of Products Levied by the Waste Electrical Appliances Treatment Fund" [17], "Notice on Improving Policies for the Treatment of Waste Electrical and Electronic Equipment" [18], "Guidelines for Dismantling, Disposing and Production Management of Waste Electrical and Electronic Equipment" (2015 edition) [19], "Guidelines for the Audit of the Dismantling and Treatment of Waste Electrical and Electronic Equipment" (2015 edition) [20], "Disposal Catalog of Waste Electrical and Electronic Equipment" (2014 edition) [21], "Subsidy Standards for the Waste Electrical and Electronic Equipment Treatment Fund" [22], "Administrative Measures for Eligibility Licensing for the Disposal of Waste and Discarded Electrical and Electronic Equipment" [23] and "Guidelines for the Audit of the Dismantling and Treatment of Waste Electrical and Electronic Equipment" (2019 edition) [24], etc., were issued one after another. Among them, "Measures for the Administration of the Restricted Use of the Hazardous Substances Contained in Electrical and Electronic Equipment" [25], which was announced on 6 January 2016 and implemented on 1 July 2016, proposed the management of the restricted use of hazardous substances, including the types of electrical and electronic equipment, the types of hazardous substances that are restricted, restricted use time and exemption requirements. At the same time, it clarified the qualification audit system for the restricted use of the hazardous substances in electrical and electronic equipment, which is the Chinese version of the RoHS Directive. In 2020, the "Implementation Plan on Improving the Recycling and Treatment System of Waste Home Appliances and Promoting the Renewal of Household Appliances Consumption" run through the industrial chain of recycling and treatment of WEEE [26]. In the same year, the 17th Meeting of the Standing Committee of the 13th National People's Congress reviewed and passed the revised "Law of the People's Republic of China on the Prevention and Control of Environment Pollution Caused by Solid Wastes" [27] which redefined the extended producer responsibility system, multichannel collection and centralized processing and disposal system for WEEE. It is expected that the e-waste industry will usher in new development opportunities, thereby promoting the transformation, upgrading and long-term development of the industry.

During this time, a large number of useful explorations were carried out. In 2004, the National Development and Reform Commission supported the construction of WEEE resource treatment objects in Qingdao, Hangzhou, Beijing and Tianjin with the national debt

funds. The "Home Appliances to the Rural Areas" campaign was launched in December 2007 and fully promoted in February 2009. From June 2009 to the end of 2011, the "replacement of old for new" activity for home appliances greatly increased the formal recovery rate of WEEE and promoted the development of the Chinese formal WEEE disposal system. The subsidy policy for WEEE was formally implemented in July 2012, which became the main driving force to promote the recycling and disposal of WEEE. "The Implementation Plan of the Extended Producer Responsibility System" [28] put forward clear requirements for promoting the implementation of the extended producer responsibility system for electrical and electronic equipment. At present, China has formed a comprehensive and sound legal and regulatory management system for e-waste recycling and pollution control covering the collection, transportation, treatment and disposal as well as the production, import and export, sales and other parts of electrical and electronic equipment.

## 4. Chinese WEEE Achievements in Recent Years

Under the vigorous promotion of WEEE-related policies, the Chinese e-waste recycling industry has also developed rapidly, which is mainly reflected in the following four aspects: effective containment of informal processing due to the rapid development of the WEEE processing industry, continuous improvement of management methods, significant resource and environmental benefits and the world's leading recycling efficiency.

From 2012 to 2019, a total of five batches of 109 authorized WEEE recycling companies were included in the list of WEEE fund subsidies distributed in 29 provincial administrative units, with an annual processing capacity of 163 million units. As shown in Tables 1 and 2, the amount of WEEE dismantling in China increased significantly. In 2019, 83.56 million WEEE units were recycled and disposed of, including 43.19 million TVs, 10.77 million refrigerators, 15.73 million washing machines, 6.2 million air conditioners and 7.67 million computers. Compared with 2013, the amount of dismantling in 2014 nearly doubled, and the average growth rate from 2014 to 2019 exceeded 13% [29,30]. At the same time, a set of technical routes combining labor and machinery suitable for China was developed, and the dismantling technology, equipment and production lines have been continuously standardized and improved, as shown in Figure 1.

**Table 1.** Dismantling volume of WEEE of the 109 qualified enterprises from 2013 to 2019 (10,000 units).

|      | TVs  | Refrigerators | Washing Machines | Air Conditioners | Computers | Total  |
|------|------|---------------|------------------|------------------|-----------|--------|
| 2013 | 4027 | 62            | 171              | 0.5              | 126       | 4386.5 |
| 2014 | 5764 | 158           | 328              | 11               | 785       | 7046   |
| 2015 | 5317 | 333           | 637              | 19               | 732       | 7625   |
| 2016 | 4425 | 615           | 1271             | 210              | 1492      | 8013   |
| 2017 | 4208 | 804           | 1360             | 398              | 1227      | 7997   |
| 2018 | 4253 | 922           | 1441             | 506              | 978       | 8100   |
| 2019 | 4319 | 1077          | 1573             | 620              | 767       | 8356   |

**Table 2.** WEEE dismantling rate increase in 2013–2019.

| Year                  | 2014   | 2015  | 2016  | 2017   | 2018  | 2019  | Average |
|-----------------------|--------|-------|-------|--------|-------|-------|---------|
| Yearly rate increase, % | 60.63% | 8.17% | 5.13% | −0.20% | 1.29% | 3.17% | 13.03%  |

In 2012, the qualified dismantling enterprises first adopted daily reports for the information management of their operations. In 2016, the technology of the Internet of Things and video monitoring methods were comprehensively applied to achieve full coverage, no dead ends, and real-time dynamic monitoring of the production and operation process. Third-party professional organizations have been invited to conduct independent audits of the standardized management, key dismantling products, hazardous wastes of the qualified dismantling companies every quarter. Taking Sichuan as an example, a "four-level audit mechanism" has been established, that is, enterprise self-inspection,

joint preliminary review by the local environment department and a third-party agency, audits by the municipal environment departments and cross-review and re-review by the provincial environment departments, as shown in Figure 2. At present, supervision of the qualified dismantling enterprises by governments at all levels has also shifted from the previous step-by-step management process strictly in accordance with the guidelines to the environmental risk management of hazardous waste and other emissions, which is specifically reflected in the newly revised "Guidelines for the Audit of the Dismantling and Treatment of Waste Electrical and Electronic Equipment" (2019 edition).

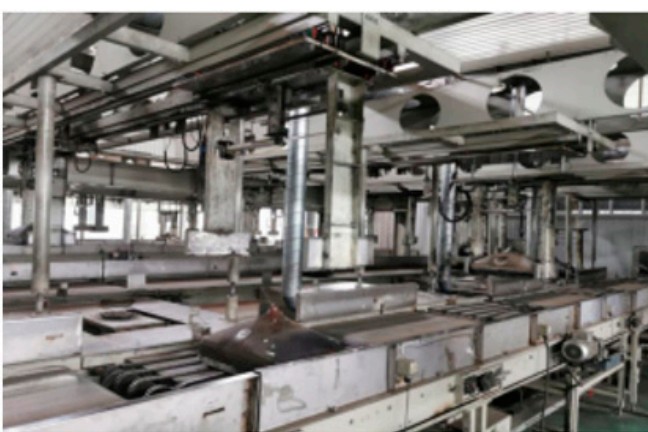
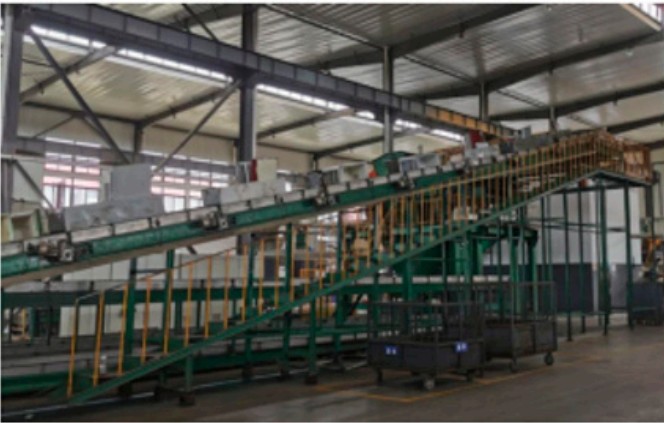
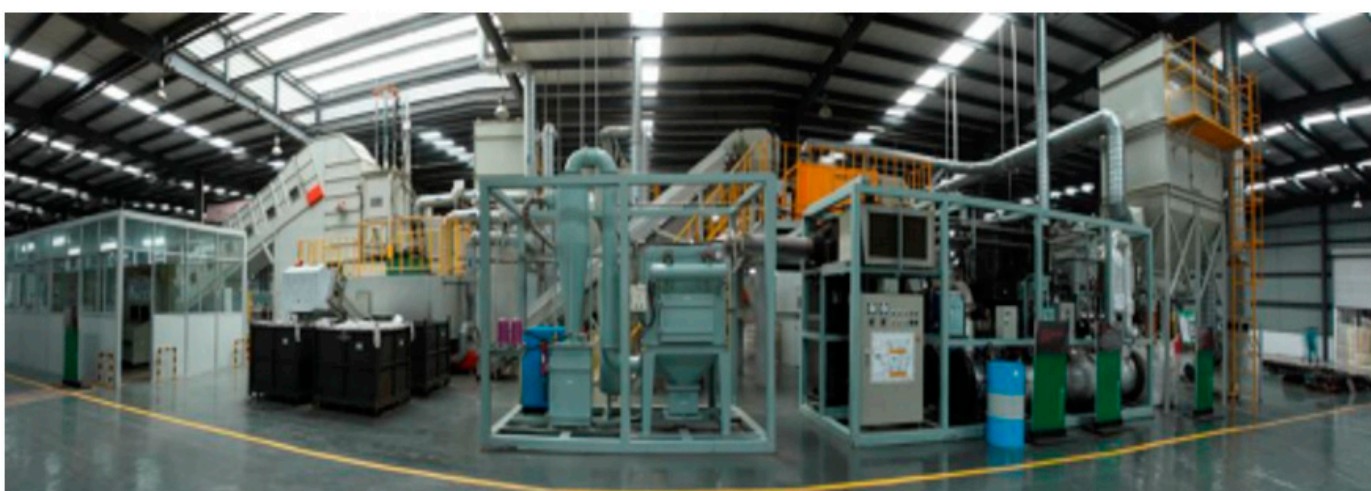

**Figure 1.** Disassembling line of an e-waste recycling company in Sichuan.

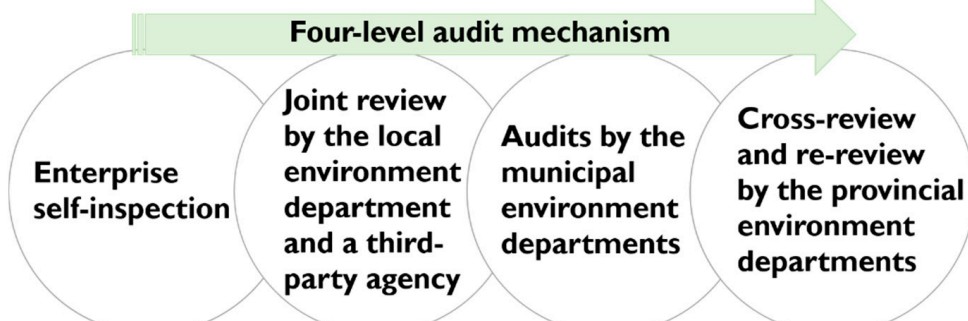

**Figure 2.** "Four-level audit mechanism" implemented by multiple departments in Sichuan.

From 2013 to 2019, a total of 12,060,000.04 tons of the key dismantling products were recycled by the 109 qualified WEEE dismantling enterprises, including 2,437,300 tons of synthetic pig iron, 375,400 tons of recycled copper, 229,400 tons of recycled aluminum and 2,404,300 tons of recycled plastics, as shown in Figure 3 [29,30]. Meanwhile, under the promotion of the "Implementation Plan on Improving the Recycling and Treatment System of Waste Home Appliances and Promoting the Renewal of Household Appliances Consumption", the Chinese WEEE recycling industry is transforming from simple dismantling to fine and high-value automation (Figure 1).

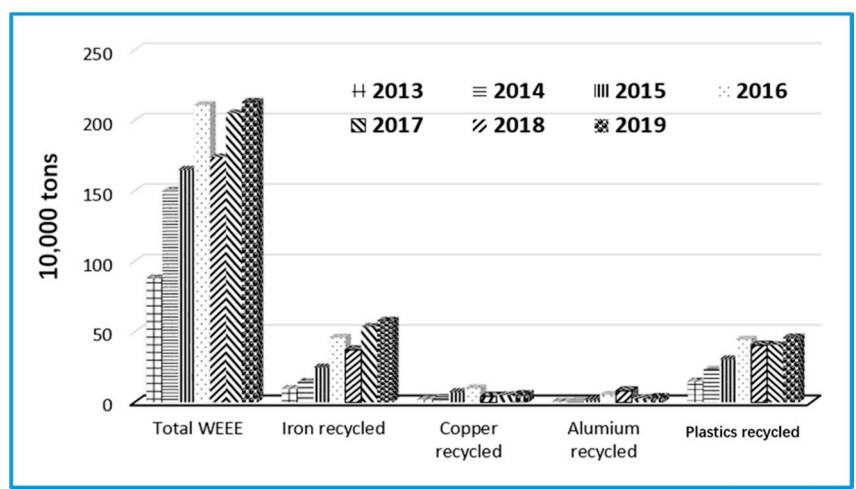

**Figure 3.** E-waste and dismantling products recycled by the qualified e-waste recycling companies.

WEEE recycling shows remarkable benefits, leads the development of the Chinese "urban mining" industry and greatly promotes development of "zero-waste cities" in China. In addition, due to the "Regulation on the Administration of the Recovery and Disposal of Waste Electrical and Electronic Equipment" and the relevant supporting policies, the dismantling activities of the 109 processing enterprises are forced to be standardized and their production management has to be carried out according to the requirements of "the fund". In the end, the standardization level of these 109 qualification enterprises is continuously improving.

## 5. Four Levels of Circular E-Waste Reutilization

Previously, scholars divided circular resource reutilization into four levels: repair, reuse and remanufacture (3Re), waste-to-materials, waste-to-products and waste-to-energy. At present, the 109 qualified dismantling enterprises recycle e-waste for the purpose of obtaining renewable resources, which belongs to the second level of resource recycling [10].

Repair, reuse and remanufacturing more or less include refurbishment (repairing products that do not meet the quality standards before sales) of the waste parts or components such as electronic components that have been repaired and restored to functional use before being consumed or used for the preparation of new products. Apple sells certified refurbished electric and electronic products at far lower prices than the new ones. The most representative example of 3Re is the refilling of glass beer bottles which has a history of more than 100 years. However, in the field of electrical and electronic products/equipment, 3Re still has a long way to go, although 3Re of WEEE and its parts/components has already formed a certain market in China with a clear division of labor and operation norms at different levels. One of the reasons is that the laws, regulations and industry-related standards relating to WEEE and its parts or components have not yet been enacted, remaining a grey area. In addition, it is worth noting that intellectual property rights of 3Re WEEE are still unresolved both at home and abroad. The ink cartridges case of Canon, Inc. vs. Recycle Assist was so serious that any company that attempts to collect unusable or discarded electronic information products for selling in the market through 3Re has to face the risk

of legal action [11]. However, according to the "Circular Economy Promotion Law of the People's Republic of China" [31] and the "Administrative Measures for the Circulation of the Used Electrical and Electronic Products" [32], 3Re WEEE, if it provides details on 3Re in a prominent position, meets quality standards and ensures traceability, could be sold without the permission of the trademark owner of the original product and is not an infringement. However, in China, most of these activities are personal endeavors, while corporate operations are rare and mostly secretive.

According to the "Guidelines for Dismantling, Disposing and Production Management of Waste Electrical and Electronic Equipment" (2015 edition) for the recycling of e-waste by the 109 qualified enterprises in China, their recycling operations all belong to the second level, material regeneration. In other words, WEEE is regenerated into various raw materials, such as recycled copper, iron, aluminum, plastics, etc., for new products by physical and chemical means, i.e., using the mechanical physical approach adopted by most of the 109 enterprises and pyrometallurgy adopted by CECEP (SHANTOU) Recycling Resources Technology Co., Ltd.

Material transformation means that after 3Re and high-value resources extraction, the remaining part of solid waste, such as WEEE, is transformed into other materials or products via physical, chemical, microbiological and/or other means. Of course, it also includes direct transformation if it is of little value. For example, cement kilns, which China is vigorously using now to promote the co-processing of hazardous waste, was actually widely used for the treatment of coal slag and steel slag before. Building materials is recognized as the main direction for bulk solid waste consumption and the most important way for the third level of waste recycling. For electronic waste, because of its unique characteristics, there are relatively few studies and applications at the third level; a typical example is waste resin powder, generated by mechanical and physical methods for copper recycling from waste printed circuit boards, which is used as a wood–plastic material with many successful industrial application cases. However, other solutions to such low-value-added WEEE need to be further explored.

Energy recovery, in fact, is to convert the combustible components into heat energy through combustion. Then, energy is used through a heat energy conversion device. The most typical example is power generation from municipal solid waste. In terms of WEEE, most of the researches in this area focus on waste resin powder after the mechanical–physical recovery of copper from waste printed circuit boards. Because these waste resin powders contain a large amount of persistent organic pollutants such as brominated flame retardants, currently, there are no such industrial incineration treatment and disposal facilities. Meanwhile, waste polyurethane foam is in the same position as the waste resin powders, widely discussed for energy recovery but currently with no industrial application.

As mentioned previously, the recycling of WEEE in China mainly focuses on the second level but not the other three levels, especially the first level. Although the first level can greatly extend the life span of electronic and electrical equipment, significantly reduce the consumption of energy materials and provide great social, economic and environmental benefits, there is still a lot of work to be done for its large-scale promotion due to the extremely complex intellectual property and the logistics tracking of 3Re products. There are also issues of environmental risk control since the more profound the recycling of resources, the higher the environmental risk may be. Besides, there remains an environmental risk transfer during the process of resources recycling. For example, waste resin powder could be used to treat park benches. In this process, hazardous waste, waste resin powder, is converted to new products, tremendous changes in its nature. Therefore, it should be carefully and deeply considered if this process is free of environmental risk transfer.

## 6. Conclusions

Recently, China has obtained great achievements in WEEE recycling and pollution control: a significant increase in the dismantling numbers and technology, continuous improvement of the WEEE equipment and information management and significant so-

cial, economic and environmental benefits of the recycled resources. The Chinese WEEE management is turning to intelligent management of emissions supervision, especially for hazardous waste. The WEEE recycling technology is also aiming towards high-value-added and automated reutilization. It is still worth noting that the Chinese WEEE industry remains in the third level, while other levels, especially 3Re, are seriously inadequate. In-depth recycling technology remains in the exploratory stage, and WEEE recycling industry transformation and upgrading are also imminent. In addition, there is still a lot of work to be done on intellectual property, environmental risk control and transfer in the process of resource recycling.

**Author Contributions:** Conceptualization, M.C.; data curation, Y.D. and S.L.; funding acquisition, M.C.; investigation, R.W., K.Y., Y.L., M.S., J.W. and Q.L.; methodology, J.S.; project administration, M.C.; supervision, J.S. and Z.S.; writing—original draft, R.W.; writing—review & editing M.C. All authors have read and agreed to the published version of the manuscript.

**Funding:** This research was funded by National Natural Science Foundation of China (51974262), the Science & Technology Pillar Program of Sichuan Province (2019YFS0450).

**Conflicts of Interest:** The authors declare no conflict of interest.

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
