# Peer review of "Waste Electrical and Electronic Equipment Reutilization in China"

_sustainability, doi:10.3390/su132011433_

Round 1
Reviewer 1 Report
The manuscript “Primary Introduction of Waste Electrical and Electronic Equipment Circular Reutilization in China” by authors Rong Wang, Yi Deng, Shuyuan Li, Keli Yu, Yi Liu, Min Shang, Jiqin Wang, Jiancheng Shu, Zhi Sun, Mengjun Chen and Qian Liang, can be published after the following significant additions and corrections:
- I ask the authors to clearly emphasize in the introductory part of the paper why WEEE recycling is important (the fastest growing waste stream in the world, they are the main source of secondary raw materials of precious metals, importance from an economic point of view, potential environmental impacts,…),
- Also, in the introductory part emphasize why WEEE recycling is a complex and demanding process (diversity of constituent materials…),
- The Abstract states that China has made great achievements in WEEE circular reutilization and pollution control! Solving the biggest problem in the field of e-waste recycling refers to the technological complexity of the process of valorization of useful components from this type of waste. In accordance with the above, I ask the authors to give an overview of the state of the art in the introductory part of the paper through a review of scientific and patent literature with a brief analysis of the advantages of individual procedures. Also, within the manuscript it is necessary to give a brief overview of representative procedures for selectively obtaining useful components contained in e-waste, which are applied in China.
Author Response
- General Comments
The manuscript “Primary Introduction of Waste Electrical and Electronic Equipment Circular Reutilization in China” by authors Rong Wang, Yi Deng, Shuyuan Li, Keli Yu, Yi Liu, Min Shang, Jiqin Wang, Jiancheng Shu, Zhi Sun, Mengjun Chen and Qian Liang, can be published after the following significant additions and corrections.
Response
We thank the reviewer for the positive comments on this manuscript.
- I ask the authors to clearly emphasize in the introductory part of the paper why WEEE recycling is important (the fastest growing waste stream in the world, they are the main source of secondary raw materials of precious metals, importance from an economic point of view, potential environmental impacts,…). Also, in the introductory part emphasize why WEEE recycling is a complex and demanding process (diversity of constituent materials…).
Response
Thanks for this suggestion. This part has been revised as follow.
E-waste, or electronic waste, with a formal name of waste electric and electronic equipment (WEEE), is attracting wide attention from both China and the world. As the fastest growing waste stream in the world, it is the main source of secondary raw materials of resources, such as Cu, and precious metals like Pd, Pt, Au and Ag [1]. From an economic point of view, WEEE has a higher economic value than their original ores [2]. However, it also contains potential hazardous to the eco-environment and human health due to the high concentrations of heavy metals and organic pollutants [3]. The diversity of the constituent materials, high value but hazard, make the recycling of WEEE became a complicated and demanding process.
- The Abstract states that China has made great achievements in WEEE circular reutilization and pollution control! Solving the biggest problem in the field of e-waste recycling refers to the technological complexity of the process of valorization of useful components from this type of waste. In accordance with the above, I ask the authors to give an overview of the state of the art in the introductory part of the paper through a review of scientific and patent literature with a brief analysis of the advantages of individual procedures. Also, within the manuscript it is necessary to give a brief overview of representative procedures for selectively obtaining useful components contained in e-waste, which are applied in China.
Response
Thanks for the suggestion. It could be a great ideal for review the state of the art of scientific and patent literature for introducing the achievement for WEEE in China. Here, in this manuscript, we simply introduced the achievements of WEEE in China in section 4, from 4 aspects: effective containment of informal processing due to the rapid development of WEEE processing industry, continuous improvement of management methods, significant resource and environmental benefits, and an international leading recycling efficiency. We will process this review in the near future.
We thank all the reviewers for the constructive comments!
Reviewer 2 Report
The conclusions could be improved by the authors.
Author Response
- The conclusions could be improved by the authors.
Response
Thanks for the suggestion. We changed the title of this section from conclusions and perspective to conclusions. It has been revised to “Recently, China has made great achievements in WEEE recycling and pollution control: a significant rising in dismantling number and dismantling technology, continuously improving in WEEE equipment and information management, and significant social, economic and environmental benefits from the recycled resources. Chinese WEEE management is turning to intelligent management of emissions supervision, especially for hazardous waste. WEEE recycling technology is also aiming to high-value added and automated reutilization. It is still worth to be noted that Chinese WEEE industry remains in the third level while other levels, especially 3Re, are seriously inadequate. In-depth recycling technology remains in the exploratory stage, and WEEE recycling industry transformation and upgrading is also imminent. In addition, there is still a lot of work to be done on intellectual property, environmental risk control and transfer during the process of resource recycling.”
We thank all the reviewers for the constructive comments!
Reviewer 3 Report
Interesting work done on this particular area providing information to people related to this topic.
I would suggest the following aspects to be considered:
Title: “Primary Introduction of Waste Electrical and Electronic Equipment Circular Reutilization in China”. The title is not representative of the content of this manuscript. In the manuscript there is no discussion on the circular reutilization of e-wastes. I would suggest: “Waste Electrical and Electronic Equipment Reutilization in China”.
Abstract: Writing issues. “…For the perspectives, WEEE management will turn to intelligent management targeted on hazardous wastes and other pollutions but not merely the guidelines, while its technology will transfer to value-added and automated reutilization targeted on more specific but not just simple dismantling…”. Please rephrase this sentence as it is not understandable at all. Try to break it down into several understandable sentences
Abstract: “…Meanwhile, we analyzed the four level of WEEE circular recycle: repair, reuse and remanufacture, waste to materials, waste to products and waste to energy, and also put forward the questions during this process…”. Repair, reuse and remanufacture were indeed discussed in the paper but there was no debate about “waste to materials, waste to products and waste to energy”. I highly recommend to explain this situation.
Introduction: define ordinary people.
Introduction: “…Then, we deeply discussed the probl7ems…”. Correct errors
Definition: “…elements/devices and materials, etc…. ». Please delete “etc” or mention that this term represents.
Chinese WEEE Management policy: “…recycling of non-ferrous metals and precious metals from WEEE by primitive disposal methods, such as open burning, etc…”. Please mention the other disposal methods with a suitable reference. This information is crucial for readers.
Chinese WEEE Management policy: “…the National Development and Reform Commission supported the construction of WEEE resource treatment projects in Qingdao, Hangzhou, Beijing and Tianjin with the funds of national debt…”. What kind of treatments projects were supported. Mention names and references.
Legends to figures: The captions of figures and tables must contain all the information necessary to facilitate understanding, without having to refer to the main text of the manuscript. For example: Figure 2. “ Four-level audit mechanism in Sichuan”. This legend has no precise information. Authors has to write down more information about.
Second line, page 6: “…At eh end…”. What is the meaning of this expression?
Almost at the end of page 6: “…But in Chia, most of these…”. Chia ?
Page 7: “…me[1]chanical-physical recovery of cooper from waste…”. What is cooper?
Page 7: “…widely discussed for energy recovery but currently no indus[1]trial application…”. Add “with” before “no industrial application”.
Author Response
- 1. General comments
Interesting work done on this particular area providing information to people related to this topic.
Response
We thank the reviewer for the positive comments.
- Title: “Primary Introduction of Waste Electrical and Electronic Equipment Circular Reutilization in China”. The title is not representative of the content of this manuscript. In the manuscript there is no discussion on the circular reutilization of e-wastes. I would suggest: “Waste Electrical and Electronic Equipment Reutilization in China”.
Response
Thanks for this suggestion. We have changed the title to “Waste Electrical and Electronic Equipment Reutilization in China”.
- Abstract: Writing issues. “…For the perspectives, WEEE management will turn to intelligent management targeted on hazardous wastes and other pollutions but not merely the guidelines, while its technology will transfer to value-added and automated reutilization targeted on more specific but not just simple dismantling…”. Please rephrase this sentence as it is not understandable at all. Try to break it down into several understandable sentences
Response
Thanks for the suggestion. We have rewritten the sentence to “For the perspectives, WEEE management will turn to intelligent management targeted on hazardous wastes and other pollutions but not merely the guidelines. Meanwhile, WEEE technology will transfer to value-added and automated reutilization but not just simple dismantling.”
- Abstract: “…Meanwhile, we analyzed the four level of WEEE circular recycle: repair, reuse and remanufacture, waste to materials, waste to products and waste to energy, and also put forward the questions during this process…”. Repair, reuse and remanufacture were indeed discussed in the paper but there was no debate about “waste to materials, waste to products and waste to energy”. I highly recommend to explain this situation.
Response
Actually, we have discussed “waste to materials, waste to products and waste to energy”. The “waste to materials” is discussed the last paragraph on page 6, the “waste to products” is shown the first paragraph on page 7, and the “waste to energy” could be find in the second paragraph on page 7.
- Introduction: define ordinary people.
Response
The ordinary people has been defined and revised to “people who are dismantling and using WEEE in individual workshops”.
- Introduction: “…Then, we deeply discussed the probl7ems…”. Correct errors
Response
It has been revised.
- Definition: “…elements/devices and materials, etc…. ». Please delete “etc” or mention that this term represents.
Response
Thanks for the suggestion, and we have deleted “etc”.
- Chinese WEEE Management policy: “…recycling of non-ferrous metals and precious metals from WEEE by primitive disposal methods, such as open burning, etc…”. Please mention the other disposal methods with a suitable reference. This information is crucial for readers.
Response
In Guiyu, primitive and crude techniques were used, including (1) dismantling electronic equipment; (2) heating and manual removal of components from printed circuit boards; (3) opening burning cables and wires for recovering metals; (4) chipping and melting plastics; (5) toner sweeping; (6) open acid leaching of e-waste to recover precious metals. These techniques have released a large quantity of toxic heavy metals and organic pollutants into the workplace and the surrounding environment [4].
- Chinese WEEE Management policy: “…the National Development and Reform Commission supported the construction of WEEE resource treatment projects in Qingdao, Hangzhou, Beijing and Tianjin with the funds of national debt…”. What kind of treatments projects were supported. Mention names and references.
Response
There is no detailed information about the projects. Years ago, the Chinese government carried out a large number of useful explorations on WEEE disposal. The above cities are pilot cities in one of the exploratory stages.
- Legends to figures: The captions of figures and tables must contain all the information necessary to facilitate understanding, without having to refer to the main text of the manuscript. For example: Figure 2. “Four-level audit mechanism in Sichuan”. This legend has no precise information. Authors has to write down more information about.
Response
Thanks for the suggestion. We have revised the caption of Figure 2. “Four-level audit mechanism in Sichuan” to Figure 2. “Four-level audit mechanism” implemented by multiple departments in Sichuan.
- Second line, page 6: “…At eh end…”. What is the meaning of this expression?
Response
It has been revised to “…At the end…”.
- Almost at the end of page 6: “…But in Chia, most of these…”. Chia ?
Response
It has been revised to “China”.
- Page 7: “…me[1]chanical-physical recovery of cooper from waste…”. What is cooper?
Response
It has been revised to “copper”.
- Page 7: “…widely discussed for energy recovery but currently no indus[1]trial application…”. Add “with” before “no industrial application”.
Response
It has been revised according to the suggestion.
We thank all the reviewers for the constructive comments!
- Wei, L. and Y. Liu, Present Status of e-waste Disposal and Recycling in China. Procedia Environmental Sciences, 2012. 16: p. 506-514.
Round 2
Reviewer 1 Report
I have no more comments.
Reviewer 3 Report
None